# LEARNING DYNAMIC REPRESENTATIONS OF THE FUNCTIONAL CONNECTOME IN NEUROBIOLOGICAL NETWORKS

**Luciano Dyballa**[1]**, Samuel Lang**[2]**, Alexandra Haslund-Gourley**[1]**,**
**Eviatar Yemini**[2*]**, Steven W. Zucker**[1*]
[1]Dept. Computer Science, Yale University; [2]Dept. Neurobiology, UMass Chan Medical School

## ABSTRACT

The static synaptic connectivity of neuronal circuits stands in direct contrast to the dynamics of their function. As in changing community interactions, different neurons can participate actively in various combinations to effect behaviors at different times. We introduce an unsupervised approach to learn the dynamic affinities between neurons in live, behaving animals, and to reveal which communities form among neurons at different times. The inference occurs in two major steps.[1] First, pairwise non-linear affinities between neuronal traces from brain-wide calcium activity are organized by non-negative tensor factorization (NTF). Each factor specifies which groups of neurons are most likely interacting for an inferred interval in time, and for which animals. Finally, a generative model that allows for weighted community detection is applied to the functional motifs produced by NTF to reveal a dynamic functional connectome. Since time codes the different experimental variables (e.g., application of chemical stimuli), this provides an atlas of neural motifs active during separate stages of an experiment (e.g., stimulus application or spontaneous behaviors). Results from our analysis are experimentally validated, confirming that our method is able to robustly predict causal interactions between neurons to generate behavior.

## 1 INTRODUCTION

The connectome in neurobiology might seem roughly analogous to the architecture of artificial neural networks (ANNs) in artificial intelligence (AI), in the sense that it specifies the relevant structure of connections. In AI, engineering this architecture is an engaging part of problem formulation; in neurobiology, the structure of the connectome complicates the process of circuit dissection for several reasons. First, there are different types of connections between neurons: some are due to chemical synapses or electrical ones (gap junctions) (White et al., 1986; Jarrell et al., 2012; Cook et al., 2019; Witvliet et al., 2021), and others induced by, e.g., extrasynaptic signaling of monoamines or neuropeptides (Bentley et al., 2016; Ripoll-Sánchez et al., 2022; Beets et al., 2023). Each one defines a distinct connectome (Fig. 1a). Secondly, unlike ANNs, in biology different connections can be engaged at any given moment (e.g., as a result of neuromodulatory signals that alter system-wide states (Bargmann & Marder, 2013)). As in complex social networks, the effective—or functional—connectome varies with task and animal state; these dynamics render the predictive power of the biological connectomes problematic. We agree with others working in different domains, e.g. Bassett et al. (2011); Skarding et al. (2021), that a functional connectome that is dynamic across time is required, given that a number of phenomena have been observed wherein neurons flexibly change how they encode behavior in a state-dependent manner (Atanas et al., 2023). Our goal is to learn a representation of these time-varying communities of neurons organized by behavioral responses in a "simple" organism, *Caenorhabditis elegans*.

We present a novel algorithm for learning the dynamic community organization within a weighted connectome, based on brain-wide activity measurements of each individual neuron. Our main con-

---

*Correspondence to: `eviatar.yemini@umassmed.edu`, `steven.zucker@yale.edu`.

[1]Code is available at https://github.com/dyballa/dynamic-connectomes.

tribution is to organize the connections—or estimated similarities between neural activity—rather than the neurons themselves. It differs from previous attempts by considering the full system of similarities and animals across time, rather than taking a step-by-step statistical approach (Varshney et al., 2011; Bentley et al., 2016; Cook et al., 2019; Uzel et al., 2022). Our approach combines two previously unrelated components: (*i*) the use of tensor factorization to reveal putative groups of related neurons/animals in time, which are then (*ii*) passed to a community detection algorithm. Although the emphasis here is on the algorithm, our results are confirmed with experiments that silence specific neurons predicted by our method to directly measure their impact on behavior. Moreover, even though the dynamic functional connectomes we reveal are specific to *C. elegans*, we believe our approach is widely applicable to learning representations of dynamic communities of neurons in other organisms, as well as in other domains, such as fMRI and social networks.

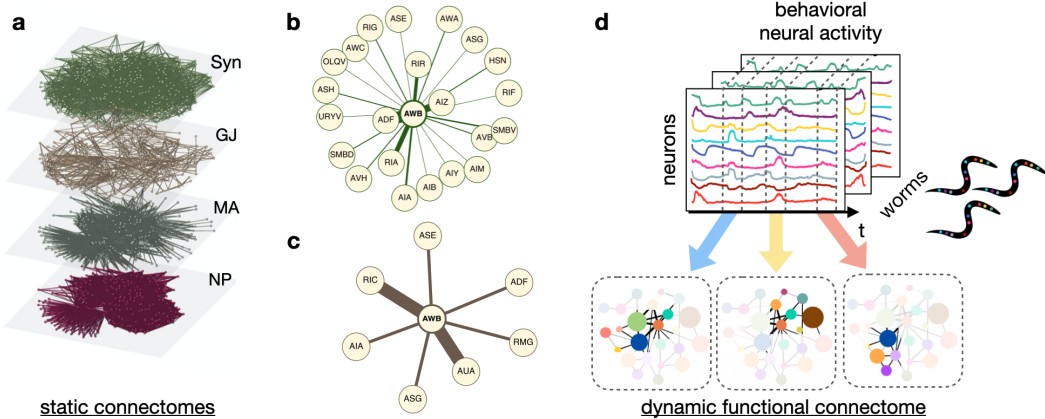

Figure 1: From static to dynamic connectomes. (**a**) Different modes of neural communication are represented by substantially different connectomes. Shown here are chemical synapses (Syn), gap junctions (GJ), monoamines (MA), and neuropeptide (NP) connectomes for the nematode worm *C. elegans* (image adapted from Bentley et al. (2016)). How these relate to behavior has been problematic, however. The immediate neighborhoods for neuron AWB are shown (**b**) from Syn and (**c**) from GJ; which governs AWB communication and when? (**d**, *top*) The dataset consists of activity traces from each neuron in *C. elegans* across time; dashed lines illustrate stimulus presentation of e.g. a repulsive chemical that is directly sensed by the worm. Note how the traces differ across time and across worms. (**d**, *bottom*) Different modules, or communities of neurons, become active at different time periods, revealing how neurons interact dynamically to encode behavior. Our goal is to infer these communities comprising the dynamic functional connectome.

## 2 METHODS

It is now possible, using the "NeuroPAL" method (Yemini et al., 2021), to determine both the cell-type-identity of every neuron and also record its activity in *C. elegans* across time in awake, and even behaving animals (Dag et al., 2023). Previously, neural activity traces from multiple worms have been analyzed using traditional statistics such as correlation (e.g., Kato et al. (2015); Yemini et al. (2021); Susoy et al. (2021); Randi et al. (2023)), small local motifs (Uzel et al., 2022), or more complex predictive models (Cecere et al., 2021; Dag et al., 2023) which assume that most neurons will respond in roughly the same way when exposed to the same stimulus. However, both past and recent studies indicate different ways that neurons can change how they encode behavior in a state-dependent manner (Bargmann & Marder, 2013; Gordus et al., 2015; Atanas et al., 2023), despite identical repeated presentations of chemosensory stimuli intended to produce stereotypical behavior. Multiple worms are therefore needed to determine whether "typical" functional networks might activate after the introduction of a particular stimulus.

The third dimension (individual worms) in the data suggests using a tensor rather than a matrix for representation of neural activity (Williams et al., 2018; Dyballa et al., 2024). A naive tensor formulation would be to build it using the temporal activity directly: WORMS × NEURONS × TEMPORAL

TRACES (e.g., Williams et al. (2018)). However, this would restrict the similarity computation between traces to a multi-linear form, which can be ambiguous for imaging data. We propose instead a novel tensor formulation with dimensions: TIME × WORMS × PAIRWISE AFFINITIES, where instantaneous affinities (that may vary over time) are precomputed from the temporal traces of each pair of neurons. In this way, non-linearities can be introduced by performing the affinity computations between neurons separately from the factorization (see below). Moreover, given that neural affinities are expected to be state-dependent and changing over time, the tensor components will be able to provide a summarized description of which communities or sub-networks are active and when. We demonstrate this in Results.

## 2.1 INTERPRETABLE, INSTANTANEOUS AFFINITIES; FIG. 3A

Experimental recordings of *C. elegans* currently span several minutes to nearly half an hour in duration, during which it is likely that different groups of neurons will become active at different times. Affinities should thus be a function of time, rather than the more traditional method of computing a global-time measure such as correlation (e.g., Yemini et al. (2021); Randi et al. (2023)). We compute, instead, local non-linear similarities between pairs of neural trace curves, which we term differential affinity.

Building a time-varying similarity measure between traces is a subtle problem. For example, two neurons that are mostly silent throughout an experiment will have small Euclidean distance (and high cosine similarity) between them, even though there may be no evidence that the neurons participate in the same circuit. Or, if they happen to maintain high, constant levels of activity at some point in time, then their distances will also be locally small, but if they reached those levels several seconds apart, it is unlikely that the two neurons are actually interacting. The instantaneous trace values are not meaningful individually: the temporal history of how they arrived at that level is highly relevant. We rely on the information present in the rates of monotonic change in activity, and compute what we term *local differential affinity*.

We compare two neurons' derivatives during intervals in which both had a constant sign, i.e., periods of monotonic increase or decrease in activity. The intuition is that two neurons with coinciding changes in activity are likely to be influencing one another, or being influenced by a common phenomenon, and thereby participating in the same functional circuit. The details of this computation are given in the Appendix; examples of the instantaneous local differential affinity are shown in Fig. 2.

Affinities show how similar two curves are, locally, in terms of their absolute derivatives, which is motivated by the fact that two neurons with very similar but opposite sign derivatives are still likely to be interacting by means of some inhibitory mechanism. As a result, they can be interpreted as how likely it is for the two neurons to be interacting (either directly or indirectly, perhaps via common input). The instantaneous pairwise affinities $a_{ij}^{(t)}$ at a time $t = 1, \ldots, T$ may be bundled into an affinity matrix, $A^{(t)}$, which is used in the tensor incorporating information across multiple worms.

## 2.2 DYNAMIC AFFINITY TENSOR DECOMPOSITION; FIG. 3B

We now have a set of affinity matrices $\{A^{(t)}\}$, each one encoding an affinity network at a time $t$, on which a community detection algorithm could be run. This would yield a collection of "instantaneous" network communities. However, which instants should be selected? How long should they be? Are they independent? Rather than addressing these questions individually, we exploit the summarization power of non-negative tensor factorization to automatically cluster affinity networks across time (and animals!): we seek contiguous intervals during which a given affinity pattern is approximately preserved across several worms. Recall that our tensor has dimensions: TIME × WORMS × PAIRWISE AFFINITIES. (We note as an aside that this tensor can also help with "completing" affinity matrices containing missing data from a few neurons.) See Fig. 3 for a graphical illustration of the following discussion.

If there are $n$ neurons, $A(t)$ has dimension $n \times n$. Note that there is no expectation for parts of the connectivity structure in $A(t)$ to be decomposable as the product of two matrices. Therefore, instead of stacking the 2-D matrices in a 4-way tensor, we vectorize each $A(t)$ by reshaping it as a

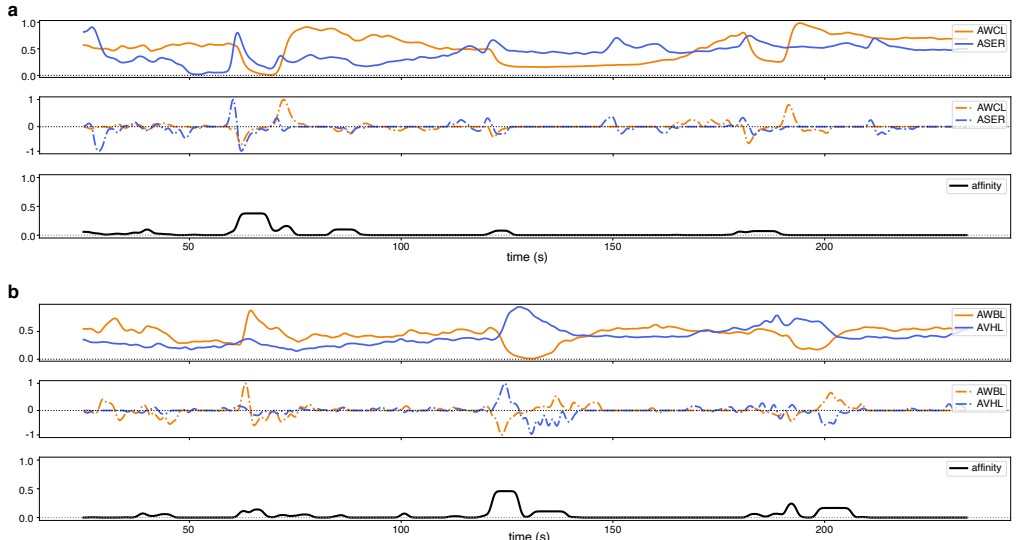

Figure 2: Computation of dynamic differential affinity. (**a**, *top*) Activity traces of two neurons, AWCL and ASER, across time. Note that they share marked periods of similar trends in activity. (**a**, *middle*) Smoothed derivatives of the top traces. Note how the derivatives agree during periods in which both traces vary together. (**a**, *bottom*) Affinity trace across time computed between AWCL and ASER (see section 2.1 and Appendix A.1 for details). Note how the "bumps" coincide with regions of potential interest. (**b**) Similar plots for a different neuron pair. Again, our non-linear affinity measure has low value except for periods when both neurons increase or decrease their activities, i.e., are more likely to be interacting.

vector of dimension $n(n-1)/2$, the total number of pairs (for computational efficiency, since they are symmetric, only the upper-triangular part is used).

In a Canonical Polyadic (CP) decomposition (Carroll & Chang, 1970; Harshman et al., 1970; Kolda & Bader, 2009), the goal is to approximate a $n$-way tensor $\boldsymbol{T} \in \mathbb{R}^{I_1 \times I_2 \times \ldots \times I_n}$ by a sum of rank-1 tensors. Letting $R$ be the number of components chosen, we may express the specific case of a 3-way tensor as (Kolda, 2006):

$$\tilde{\boldsymbol{T}} = \sum_{r=1}^{R} v_r^{(1)} \circ v_r^{(2)} \circ v_r^{(3)}, \tag{1}$$

where $\circ$ stands for the vector outer product and $X^{(k)}$ is called a *factor matrix* containing the factors $v_r^{(k)}$ as its columns. Each rank-1 tensor is thus formed by the outer product between each factor in the same component (a *component* refers to each set of associated factors). Most tensor decomposition algorithms use squared reconstruction error as objective function (Cichocki et al., 2009):

$$\min_{X^{(1)}, X^{(2)}, X^{(3)}} \frac{1}{2} \|\boldsymbol{T} - \tilde{\boldsymbol{T}}\|^2. \tag{2}$$

where $\| \cdot \|$ is the norm of the vectorized tensor.

Since the data consist of non-negative neural affinities, we adopt a non-negative tensor decomposition, also known as non-negative tensor factorization (NTF) (Bro & De Jong, 1997), which adds a non-negativity constraint $X^{(k)} \geq 0, \forall k$ to eq. 2. Non-negative factorization is popularly used for matrices (NMF), and achieves a parts-based representation with more easily-interpretable components (Lee & Seung, 1999; Cichocki et al., 2009); additionally, it has been shown to produce more stable factors, is less prone to overfitting, and has high parameter efficiency (Williams et al., 2018). We use a hierarchical alternating least squares (HALS) algorithm (Cichocki & Phan, 2009), as implemented in the `tensortools` Python library (Williams, 2024). Its efficiency when applied to real-world datasets has been compared against several other algorithms in Phan & Cichocki (2008).

Each component resulting from our tensor will be constituted by three factors: one representing a pattern $f_a$ of (vectorized) pairwise affinities; a vector $f_t$ representing periods of time where such pat-

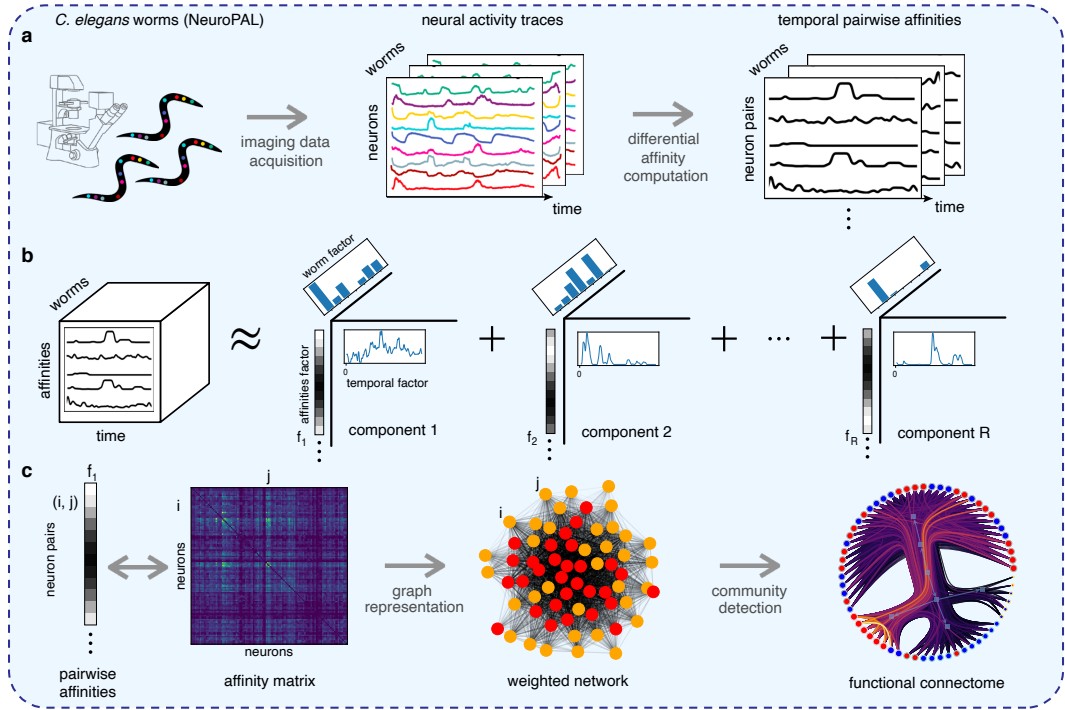

Figure 3: The procedure for inferring dynamic functional connectomes in three stages. (**a**) *Stage 1: Building affinities.* Neurons in behaving worms are imaged to yield a matrix of activity (in time) for each neuron for each worm. The differential affinity computation compares traces to yield a tensor of affinities (in time) for each neuron pair for each worm. (**b**) *Stage 2: Non-negative tensor factorization* yields components that reveal affinity patterns found for certain worms (bar plot) over different time intervals (temporal curve). For example, component 2 indicates that neuron pairs with high affinity in worms 4 and 6 are active early in the experiment, and the last factor applies mainly to worm 2. (**c**) *Stage 3: From affinities to a functional connectome.* The affinity factor from a selected tensor component is rearranged as an affinity matrix, from which an equivalent weighted graph is implicitly built. A community detection algorithm then reveals groupings of neurons behaving similarly over the indicated time interval.

tern was observed; and a vector $f_w$ whose loadings, or coefficients, indicate how much each worm's affinities can be described by $f_a$ at those times expressed in $f_t$. These components, combined, can be seen as a soft multi-clustering of worms in terms of their activity patterns and the times at which they occur. See Fig. 3b and Appendix A.4.

## 2.3 NEURAL AFFINITY COMMUNITIES; FIG. 3C

Notice that, because the neurons are sorted in an arbitrary way, looking at the affinity factor matrices should not be particularly enlightening. However, because affinities are non-negative and bounded, they can be readily treated as adjacencies (weighted edges) in a graph where the nodes correspond to individual neurons. This enables us to use a community-detection algorithm to cluster similar neurons based on their activity over time.

Although several methods for dynamic community detection (DCD) exist, using both traditional ML (Rossetti & Cazabet, 2018) and deep learning (Skarding et al., 2021), the vast majority are focused on unweighted networks (Li et al., 2021). That is because in many applications a dynamic network refers to a graph that changes its number of nodes or edges over time, not their weights (with few exceptions, e.g., Guo et al. (2014)). Of course, for our purposes the edge weights (i.e., affinity factors from NTF) are essential, as they convey the likelihood that two neurons interact over some period of time (section 2.1).

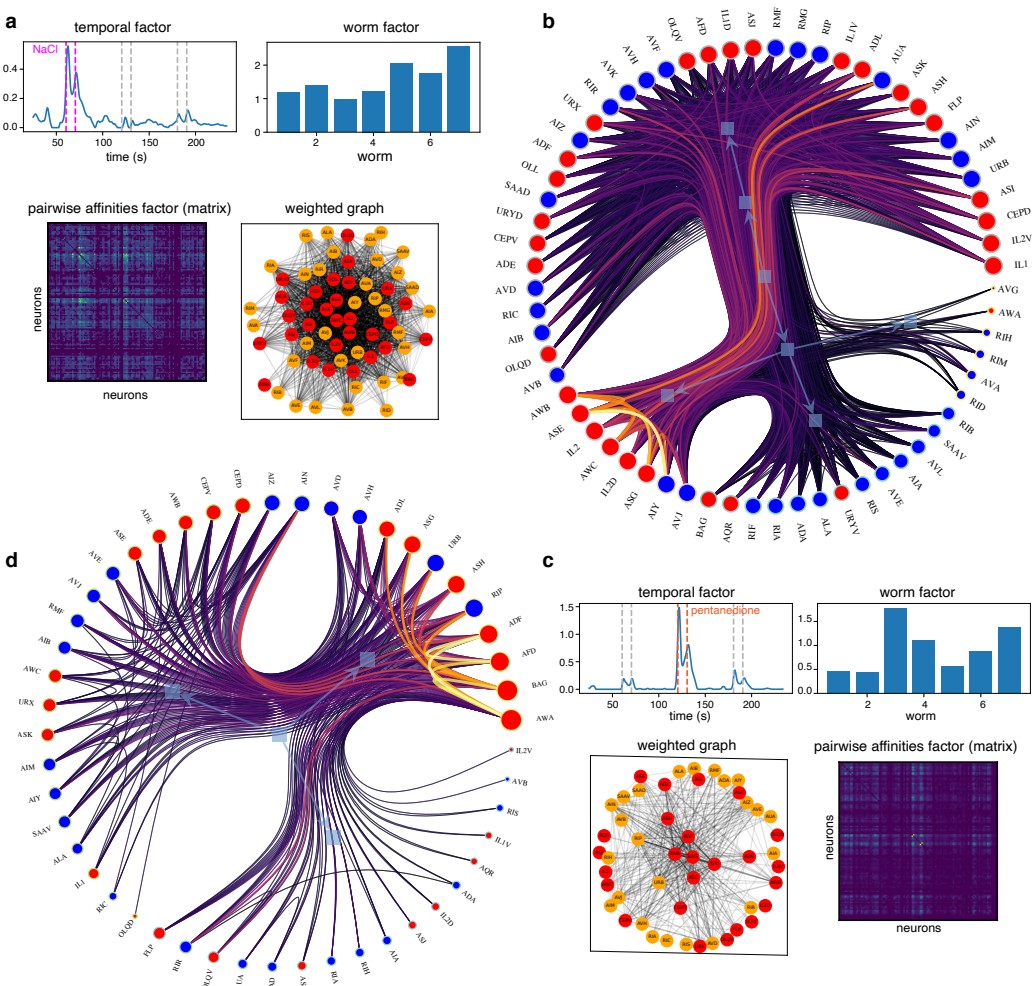

Figure 4: Tensor components for a real worm during an experiment in which repellent and attractive stimuli were applied. The application of these stimuli is important, since it provides correspondence between the time axes. (**a**) One tensor component that selects out the interval during which repulsive NaCl (salt) was applied. All worms reacted, but worms 5, 6, and 7 showed higher responses for certain pairs of neurons (from the affinity factor, shown as an unsorted matrix). The weighted graph visualized with a traditional force-directed embedding is difficult to interpret, but the nested community structure revealed by the NWSBM algorithm (**b**) provides a clear view of the neuronal modules involved in salt perception. (**c**) A different tensor component that selects several time intervals when stimuli were applied, but is especially involved with the attractant 2,3-pentanedione. Most worms are again implicated, but the community of neurons (**d**) is qualitatively different.

Since our tensor decomposition organizes the temporal dimension—i.e., which networks are active at which times—we seek a community detection method appropriate for handling edge weights. Additionally, we wish to prevent spurious aggregations of nodes due to random fluctuations in the affinity values (from possible noise in the activity traces). Therefore it is preferable to infer a latent community structure that carries explanatory power, as opposed to adopting classical descriptive methods that attempt to find communities according to some preconceived notion of a good division of the network into groups.

A classical model for community structure is the stochastic block model (SBM) (Holland et al., 1983), which groups nodes according to their probabilities of connection to the rest of the network. Statistical inference can be used to find the best-fitting model to the data. A principled approach for this task is to formulate generative models that allow this modular decomposition to be found

via statistical inference (Aicher et al., 2015; Peixoto, 2019). In particular, the nested weighted-SBM algorithm (NWSBM) (Peixoto, 2018; 2014) extends the SBM to be applied to weighted networks by treating edge weights as covariates, and employs a Bayesian approach capable of inferring the number of communities from the data in an unsupervised fashion. Moreover, it requires no hyper-parameter tuning.

For example, let the partition of the network into $C$ communities be denoted as $\boldsymbol{c} = \{c_i\}$, where $c_i \in [0, C-1]$ is the community membership of node $i$. A model can be defined that generates a network $\boldsymbol{G}$ with probability

$$P\left(\boldsymbol{G}|\boldsymbol{\theta}, \boldsymbol{c}\right), \tag{3}$$

where $\boldsymbol{\theta}$ is a set of parameters controlling how the node partition affects the structure of $\boldsymbol{G}$. Assuming there is only one choice of $\boldsymbol{\theta}$ that is compatible with the generated network, we may write the Bayesian posterior probability as

$$P\left(\boldsymbol{c}|\boldsymbol{G}\right) = \frac{P\left(\boldsymbol{G}|\boldsymbol{\theta}, \boldsymbol{c}\right) P\left(\boldsymbol{\theta}, \boldsymbol{c}\right)}{P\left(\boldsymbol{G}\right)}. \tag{4}$$

The selected model is the one found to yield the smallest description length (Peixoto, 2015), namely

$$\mathcal{L} = -\ln P(\boldsymbol{G}|\boldsymbol{\theta}, \boldsymbol{c}) - \ln P(\boldsymbol{\theta}, \boldsymbol{c}), \tag{5}$$

since eq. 4 above can be rewritten as

$$P\left(\boldsymbol{c}|\boldsymbol{G}\right) = \frac{\exp(-\mathcal{L})}{P\left(\boldsymbol{G}\right)}. \tag{6}$$

Such a problem is NP-hard in general, so the likelihood is optimized stochastically. Hence, instead of assuming that there is a single "best fit" to the model, an efficient multi-flip Markov-Chain Monte Carlo (MCMC) algorithm performs model averaging to increase the robustness of the results; additional details can be found in Peixoto (2018). This algorithm was chosen after running benchmark tests against several other popular community detection methods (section 3.3, Table 1).

## 3 RESULTS

Our approach was applied to the dataset from Yemini et al. (2021), in which calcium activity from 189 neurons in the head of *C. elegans* were recorded from a total of 21 individual worms. Each worm performed a series of behavioral tasks in response to stimuli: one gustatory repellent (NaCl) and two olfactory attractants (2-butanone and 2,3-pentanedione) were flushed into the worm surroundings over periods of 10 s. Different worms were exposed to the stimuli in 3 possible sequences. We restricted the data to sensory and inter-neurons, since those are more likely to be directly responsive to the chemosensory stimuli used.

### 3.1 DYNAMIC FUNCTIONAL CONNECTOMES

Fig. 4a,c shows informative factors resulting from running NTF on our tensor (section 2.3). Each affinity factor yields a vectorized affinity matrix, which can be reorganized as the upper-triangular part of a symmetric matrix (for ease of visualization). This matrix in turn can be interpreted as a weighted adjacent matrix of a network in which nodes represent individual neurons (section 2.3). The communities inferred by the NWSBM algorithm for each factor then apply to certain time windows and worms, weighted by their respective factor loadings, and represent a transient functional connectome. Two examples are shown in Fig. 4b,d, with their respective temporal factors revealing that such neural interactions are mostly active during the presence of NaCl and pentanedione, respectively. A richly-connected individual community from the NaCl-sensing connectome is further analyzed in Fig. 5.

### 3.2 VALIDATION EXPERIMENTS

Our algorithm predicted the involvement of multiple neurons in the salt-sensing (NaCl) circuit, as those neurons found clustered with the canonical NaCl-sensing neuron ASE. Among these predictions were several neurons not previously known to play a role in this salt-sensation circuit. For

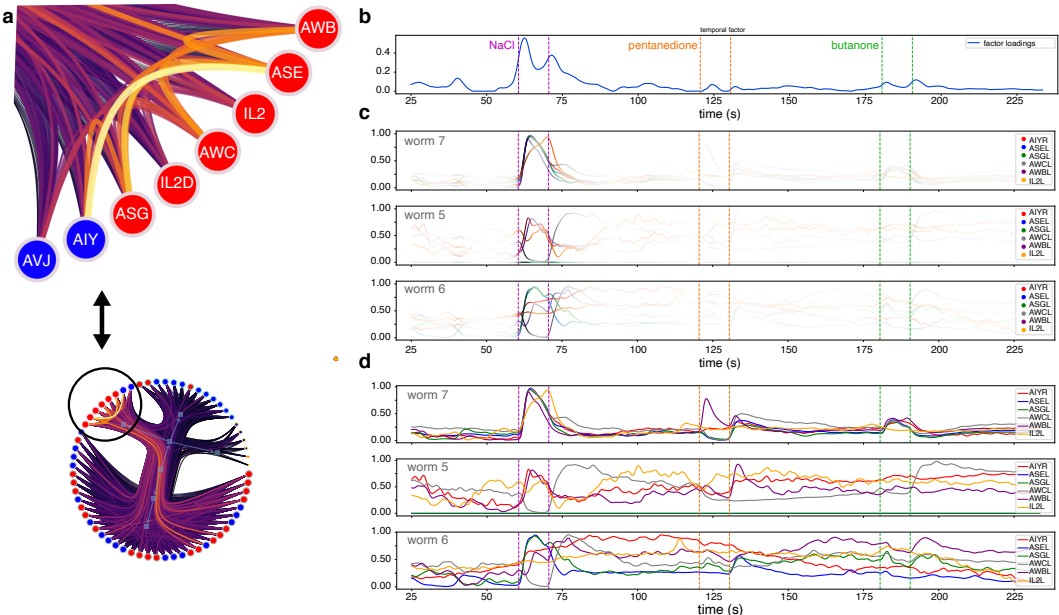

Figure 5: A transient community for salt sensing. (**a**) We here highlight one community from the dendrogram shown in Fig. 4a (cf. inset below). By reverting back to the original activity traces, we can confirm that the individual neuronal responses agree with a functional circuit organization, and that the affinity measure is meaningful. (**b**) The temporal factor of the corresponding tensor component shows that this particular circuit is most prominently involved in a response to NaCl (salt), plus minor but non-negligible responses to the other two stimuli (2-butanone and 2,3-pentanedione). (**c**) Individual activity traces for the most strongly connected neurons within the community, highlighted in proportion to the instantaneous temporal loadings. Note that the traces are most similar precisely during the NaCl interval for several worms; traces are plotted from the three worms with highest loadings in the corresponding worm factor (see Fig. 4a). (**d**) Note the complexity of the traces outside this interval. In particular, even though in worm 7 all the selected neurons were correlated throughout the experiment, this was not true for other worms, which explains the NaCl specificity in the temporal factor. Notice that, because the affinities are computed from absolute derivatives, neurons will be strongly connected even when their traces have opposite signs.

example, our algorithm predicted that the worm's primary aversive olfactory neuron AWB (Troemel et al., 1997) is substantially involved in salt sensation. Therefore, we decided to experimentally test this surprising prediction of a previously unknown role for AWB in sensing NaCl.

To do so, we compared salt avoidance in worms where AWB was functional versus worms where AWB was silenced. Briefly, we used a standard "drop test" (Bazzicalupo & Hilliard, 2005) to measure avoidance of salt, wherein a drop of 160 mM NaCl buffer (as used in Yemini et al. (2021)) was placed in front of worms, and their response was recorded as avoidance if they reversed, and non-avoidance if they continued moving forward. Each of our samples consisted of 25–50 worms that were drop tested, and collectively scored for their mean avoidance. We tested 12 such samples of functional AWB controls versus 12 samples with AWB silenced. To silence AWB, we used a published collection of worms (Wang et al., 2017), and crossed an AWB neuron-specific driver (syIs666) to an inhibitory histamine-gated chloride channel (HisCl1–syIs373) effector (Pokala et al., 2014). These AWB>HisCl1 worms (strain YYY24, with genotype syIs666;syIs373) have a functional AWB that is silenced upon treatment with histamine.

We found that silencing AWB substantially increased salt avoidance ($p$-value = 5.7e-11, mean effect size = +25%); see Fig. A2 (Appendix A.3). Our results are particularly striking since AWB is an aversive neuron and thus silencing it would be expected to decrease avoidance rather than increase it as we observed. This suggests the potential for oppositional interactions between salt sensation and olfactory aversion circuits. Our results not only validate the predictive power of our algorithm,

Table 1: Mean NMI scores of algorithms applied to the weighted-LFR benchmark

.

| Algorithm | Net 1 | Net 2 | Net 3 | Net 4 | Net 5 | Net 6 | Net 7 | Net 8 | Net 9 |
|---|---|---|---|---|---|---|---|---|---|
| NWSBM | **0.28** | **0.33** | **0.39** | **0.65** | **0.39** | **0.54** | 0.25 | 0.51 | **0.64** |
| Louvain | 0.08 | 0.19 | 0.13 | 0.04 | 0.23 | 0.29 | 0.12 | **1.0** | 0.19 |
| Combo | 0.13 | 0.18 | 0.14 | 0.04 | 0.23 | 0.28 | 0.12 | **1.0** | 0.16 |
| AS | 0.18 | 0.16 | 0.16 | 0.47 | 0.20 | 0.12 | 0.28 | 0.20 | 0.20 |
| NNSED | 0.0 | 0.18 | 0.38 | 0.32 | **0.39** | 0.50 | **0.39** | 0.43 | 0.45 |
| GNNS100 | 0.14 | 0.19 | 0.15 | 0.04 | 0.22 | 0.28 | 0.12 | **1.0** | 0.18 |

they further highlight its strength in predicting results that expert scientific researchers may find contradict assumptions they have based on published canonical neuronal roles.

## 3.3 COMPARISON WITH OTHER COMMUNITY DETECTION METHODS

The NWSBM method was compared to other popular algorithms for community detection, including classical methods such as Louvain (Blondel et al., 2008; Rossetti et al., 2019) and Combo (Sobolevsky et al., 2014), based on modularity maximization; asymptotic surprise (AS) (Traag et al., 2015); NNSED (Sun et al., 2017; Rozemberczki et al., 2020), a non-negative encoder-decoder approach; and GNNS (Sobolevsky & Belyi, 2022; Belyi, 2022), based on recurrent graph neural networks. The quality of their results was evaluated using normalized mutual information (NMI) (Danon et al., 2005), a widely adopted measure of agreement between clustering assignments (Fortunato, 2010). Because it requires ground truth—not available for the vast majority of real network datasets with weighted edges—, the algorithms above were evaluated on a benchmark of synthetic networks generated using the popular Lancichinetti-Fortunato-Radicchi (LFR) algorithm (Lancichinetti & Fortunato, 2009), a standard tool for creating benchmarks with various types of networks (Gopalan & Blei, 2013; Fortunato & Hric, 2016; Yang et al., 2016). It provides several parameters that control the how the connections and their weights will be distributed within and across communities, allowing for a rich variety of block structures to be generated. Our benchmark consisted of 9 different types of network (see Appendix A.2 and Fig. A1 for details). Table 1 shows the average NMI scores obtained by each algorithm over ten instances of each type of network. Although most of the algorithms tested was able to score highest for at least one network type, NWSBM outperformed the others in the majority of cases.

## 4 DISCUSSION AND CONCLUSION

We present here an algorithm for dynamic connectome discovery in the nematode worm *C. elegans*. The key insight was to use the differential affinity between neurons in a tensor factorization approach, rather than (the more common) use of individual neurons. The advantages were (*i*) that non-linear methods could be used in calculating affinities *before* they were tensorized; (*ii*) the temporal factor in each tensor component revealed experimental epochs during which functional circuits appeared; and (*iii*) the affinity factors could be remade into weighted graphs on which community detection algorithms could be run. In the end, the algorithm was able to make (surprising) predictions about individual neurons that were involved in unexpected functional roles, which were experimentally confirmed.

While this last point illustrates a role for machine learning in biological research, we also showed that reverting the community structure back to the original affinities (Fig. 5) can explain how the original traces caused a given pair of nodes to be ultimately grouped together.

Finally, we believe that our approach could be used more widely in understanding community behavior not only in neuroscience, but also in social and ethological situations. Working with the higher-order affinities directly informed the biology in our case; this is most likely true for other dynamical interactions as well.

ACKNOWLEDGMENTS

Supported by NIH Grant 1R01EY031059, NSF Grant 1822598.

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

# A  APPENDIX

## A.1  LOCAL DIFFERENTIAL AFFINITIES

We begin by computing each neuron $i$'s derivative $g_i(t)$ over time, pre-smoothing the raw trace $\tau_i(t)$ with a Gaussian filter of bandwidth 0.5 sec to avoid the amplification of any measurement noise (Fig. 2, top panels). Notice that periods where the derivative sign is constant must necessarily start and end at points where the derivative is either exactly zero or changes sign, and are commonly termed "zero-crossings". Because sampling is discrete over time, under a sampling interval $T$ a zero-crossing is computed as those time points $t$ such that

$$g_i(t - T/2)g_i(t + T/2) \leq 0, \tag{7}$$

i.e., there is a change of sign between two consecutive samples, and

$$|g_i(t - T/2)| + |g_i(t + T/2)| > 0, \tag{8}$$

i.e., the derivatives are not both exactly zero. When both conditions are met, we say $g_i(t)$ is a zero-crossing, and denote it as

$$g_i(t) \overset{\times}{\approx} 0. \tag{9}$$

We then partition $g_i(t)$ by those intervals where it has constant sign (i.e., those lying between zero-crossings); we call these *bumps*. Thus the support of $g_i(t)$ (subset of the time domain during which the neuron's activity is not constant), may be represented as a set of $m$ time intervals $b_i^{(1)}, b_i^{(2)}, \ldots, b_i^{(m)}$, where $b_i^{(t)}$ denotes the interval $(t_0, t_1)$, such that

$$g_i(t_0) \overset{\times}{\approx} 0 \overset{\times}{\approx} g_i(t_1). \tag{10}$$

Thus, two bumps $b_i^{(t)}$ and $b_j^{(t')}$ from neurons $i$ and $j$, respectively, will overlap when

$$b_i^{(t)} \cap b_j^{(t')} > 0. \tag{11}$$

The affinity $a_{ij}$ between neurons $i$ and $j$ at time $t$ is then defined as the fraction of overlap between the areas under their derivative bumps occurring at $t$: $b_i$ and $b_j$ respectively. This is analogous to the Jaccard index:

$$a_{ij}^{(t)} \equiv \frac{\text{area}(b_i) \cap \text{area}(b_j)}{\text{area}(b_i) \cup \text{area}(b_j)}, \tag{12}$$

where $\text{area}(b_i)$ is the unsigned area under the bump $b_i$:

$$\text{area}(b_i) \equiv \int_{t_{b_i}}^{t'_{b_i}} b_i^{(t)} dt, \tag{13}$$

with the integral taken over the time interval under each bump. Note that these affinity values lie in the range $[0, 1]$, and because they are computed locally, will be sensitive to spurious fluctuations of activity. To prevent this, we take into account global information about the traces and weight this by the relative change in trace levels during those local bumps compared to the global range of values of $\tau_i(t)$.

Notice that the areas are unsigned, meaning we ignore the signs of the bumps. This is motivated by the fact that two neurons with very similar but opposite sign derivatives are still likely to be interacting by means of some inhibitory mechanism. This further contributes to making the affinities interpretable as indicating the likelihood of interactions between pairs of neurons, regardless of the specific physiological connectivity mechanisms involved.

## A.2  WEIGHTED COMMUNITIES BENCHMARK

A total of 9 weighted networks with $N$=125 nodes (compatible with the size of our affinity networks) were generated using the weighted LFR benchmark (Lancichinetti & Fortunato, 2009). The mean neighborhood size $k$ influences the resulting number of communities produced. How such neighborhoods are distributed between intra- and inter-community connections is determined by the

topology ($\mu_t$) and weight ($\mu_w$) mixing parameters, respectively. The weights within a node's neighborhood are sampled from a power-law distribution with exponent $\beta$. Finally, the degree sequence and community sizes are drawn from power-law distributions with exponents $-\tau_1$ and $-\tau_2$, respectively. By choosing different sets of parameters for each network type, we aimed to vary several properties: the number of communities (determined by the $k/N$ ratio ); the density of inter-block connections ($\mu_w/\mu_t$ ratio); as well as the overall weight distribution ($\beta$); we thus cover a variety of scenarios. Example weighted adjacency matrices from each style and their respective parameters are shown in Fig. A1.

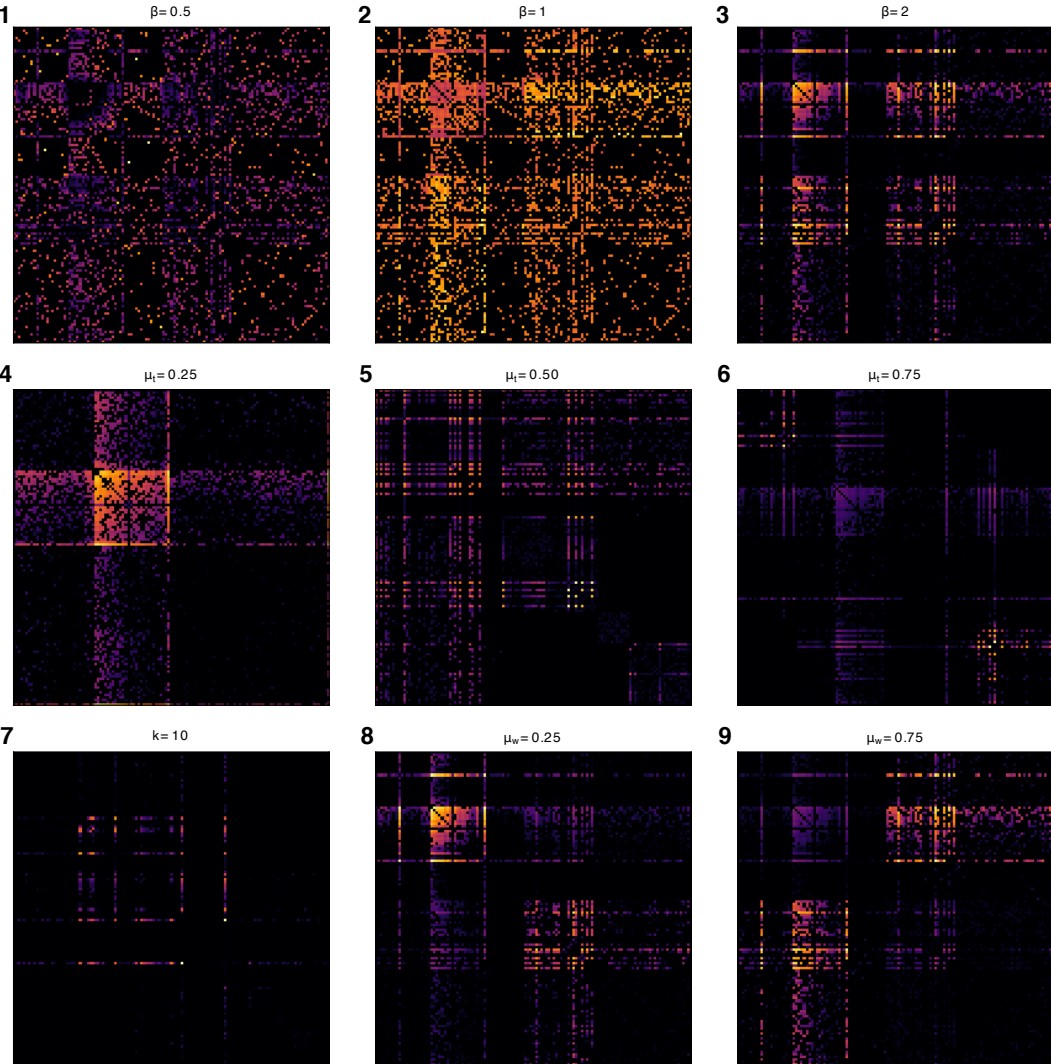

Figure A1: Family of synthetic networks generated using the weighted LFR benchmark. Each panel shows the weighted adjacency matrix of one instance of each type of network, sorted by its ground-truth community membership. Default parameter values used: $N = 125$, $k = 25$, $maxk = 100$, $\mu_w = 0.5$, $\mu_t = 0.5$, $\beta = 2$, $\tau_1 = 2$, $\tau_2 = 1$. Each network type varies a single parameter from this list (value shown above the matrix). (**1–3**) Varying $\beta$ while keeping other parameters fixed dramatically changes the weight distribution. (**4–6**) Smaller values of $\mu_t$ increase the intra-block connections (irrespective of their weights). (**7**) A smaller $k$ creates a larger number of small communities. (**8–9**) The $\mu_w/\mu_t$ ratio controls the ratio between inter- and intra-block weights.

### A.3 EXPERIMENTAL VALIDATION RESULTS

Silencing AWB substantially increased salt avoidance ($p$-value=5.7e-11, mean effect size = +25%), validating the predictive power of our algorithm; see Fig. A2. Surprisingly, this suggests the potential for oppositional interactions between salt sensation and olfactory aversion circuits.

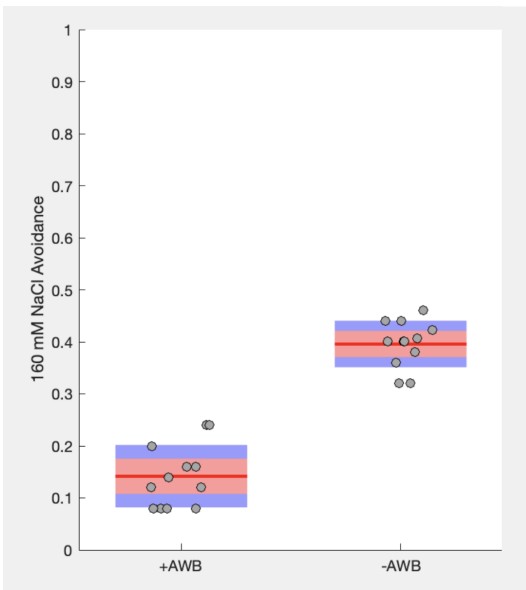

Figure A2: Results for avoidance to 160 mM NaCl for AWB controls (+AWB) vs. silenced (-AWB). Two-tailed unpaired t-test P=5.7574e-11, mean effect size is +25% avoidance when silencing AWB, N=12 vs. 12. red=95% CI, blue=1 SD.

### A.4 SELECTION OF THE NUMBER OF TENSOR COMPONENTS

Our logic for selecting the number of tensor components, $R$, was to use as many as possible to minimize reconstruction error, provided the results across multiple random initializations remained stable (i.e., small variance). Small reconstruction error suggests a faithful representation, and small variability guarantees that the retained components are robust. Based on Fig. A3a, the error variability (std. dev. across 15 runs) reaches a minimum when $R \approx$14–15 (shaded area), then increases sharply for $R > 15$. We therefore selected R=15. Fig. A3b shows that the distributions of error for each choice of $R$ are separate until $R = 15$, after which they being to overlap; this further strengthens our confidence in our choice for the number of tensor components.

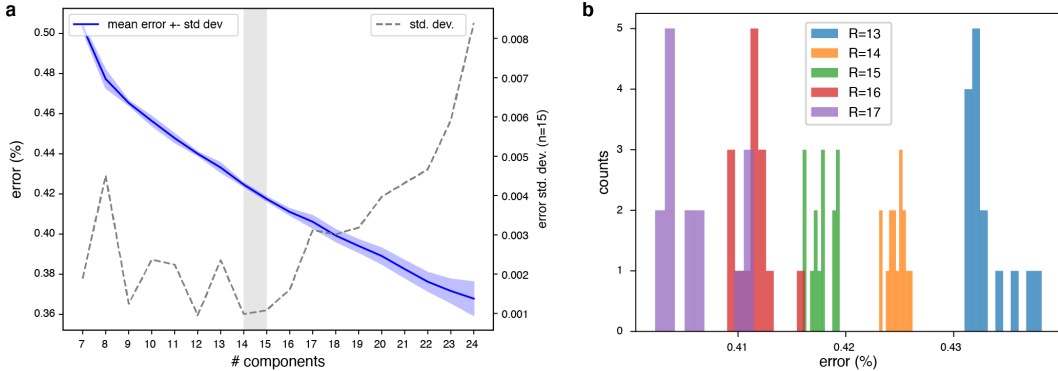

Figure A3: Selecting the number of tensor components, $R$. (**a**) Reconstruction error as a function of $R$. Although the mean error (blue curve) always decreases with increasing $R$, its variability (std. dev. across 15 runs, dashed gray curve) reaches a minimum when $R \approx 14$–$15$ (shaded area), then increases sharply for $R > 15$. (**b**) The empirical distributions of error for each choice of $R$ remain separate until $R$=15, but starting at $R = 16$ they begin to overlap. These results motivate our choice for the number of tensor components to use, namely 15.

