# OpenReview forum: "Learning dynamic representations of the functional connectome in neurobiological networks"
_ICLR.cc/2024/Conference — ICLR 2024 poster_

### Official Review · Reviewer_nAoq · 2023-10-31

**Soundness:** 3 good
**Presentation:** 3 good
**Contribution:** 3 good
**Rating:** 6
**Confidence:** 4

**Summary:**

The goal of this work is to infer community structure in neural networks (specifically that of _C. elegans_) based on functional data. To do this, the authors 1) define a pairwise affinity score computable from single-neuron time series; 2) use tensor factorization to group these pairs into dynamical motifs; 3) feed tensor components from these motifs into a community detection algorithm. When applied to data recorded from individually identified _C. elegans_ neurons, this produces networks that are tested in validation experiments, which find that perturbation of key nodes does affect network structure and behavior related to, e.g., salt avoidance.

**Strengths:**

- The problem of organizing functional data from neurons into more interpretable submodules that can be investigated in causal experiments is an important one.
- The approach is reasonable and makes use of established techniques (tensor factorization, community detection).
- Use of real neural data for both algorithm validation and suggesting perturbative experiments is a huge plus.

**Weaknesses:**

- I found the affinity score a bit _ad hoc_. I can understand the intuition, but it seems like there should be a more principled way to get at this information. Related, but along the opposite direction: why not include strong _anticorrelations_ in the affinity score? Shouldn't two neurons whose activity moves opposite to one another at most times be considered related/coupled?
- The tensor factorization will tend to treat affinities independently of one another, though the $N(N-1)/2$ affinities result from only $N$ neuronal time series. That is, the tensor factorization does not respect the underlying geometry of the problem. It's unclear to me how big an issue this is in practice, but it might lead to issues with the method.
- While the experimental data are a definite plus, it's always unclear how strongly they should be taken as validation of a particular data analysis method. In a strongly coupled network, ablating any one neuron is likely to have an effect, and it's not shown that that the method proposed here would necessarily outperform others for selecting which perturbations to apply.

**Questions:**

- How sensitive are the results presented to the particular choice of affinity score? Would, e.g., a Spearman correlation between the two time series yield qualitatively similar results?
- I might have missed this, but how did the authors decide how many tensor components to retain?

---

> ### Author Response · Authors · 2023-11-16
>
> We appreciate your time and your valuable comments and questions.
>
> **I found the affinity score a bit ad hoc. I can understand the intuition, but it seems like there should be a more principled way to get at this information. Related, but along the opposite direction: why not include strong anticorrelations in the affinity score? Shouldn't two neurons whose activity moves opposite to one another at most times be considered related/coupled?**
>
> You are absolutely correct in your intuition, and in fact that is precisely the motivation behind our proposed affinity score computation. Firstly, we use first derivatives to eliminate the influence of the raw activity trace values (which are based on the calcium fluorescence signal and may not be a fair comparison across different neurons – see Sec. 2.1, 2nd paragraph). Secondly, we use absolute values specifically to allow for neurons with opposite changes in activity to have strong affinity regardless of the derivative sign. As stated in Appendix A.1: *"Notice that the areas are unsigned (…) This is motivated by the fact that two neurons with very similar but opposite sign derivatives are still likely to be interacting by means of some inhibitory mechanism. This further contributes to making the affinities interpretable as indicating the likelihood of interactions between pairs of neurons, regardless of the specific physiological connectivity mechanisms involved."*
>
> This feature is further mentioned in the caption of Fig. 5 (last line):
>
> *"Notice that, because the affinities are computed from absolute derivatives, neurons will be strongly connected even when their traces have opposite signs."*
>
> In summary, every decision behind our affinity score has been carefully examined and extensively tested to arrive at a value that gives a reasonable measure of the likelihood that two neurons are interacting. Our experimental results further confirm the validity of our choices.
>
> Although the affinity derivation is developed in detail in Appendix A.1, based on your comments we now realize that this particular feature of our method should have been more clearly stated in the main text. So we have updated the manuscript to include in Sec. 2.1, 4th paragraph (new text in bold):
>
> *"Affinities show how similar two curves are, locally, in terms of their absolute derivatives, __which is motivated by the fact that two neurons with very similar but opposite sign derivatives are still likely to be interacting by means of some inhibitory mechanism.__ As a result, they can be interpreted as how likely it is for two neurons to be interacting"*
>
> **The tensor factorization will tend to treat affinities independently of one another, though the N(N−1)/2 affinities result from only N neuronal time series.**
>
> Although they originate from N time series, every affinity is computed independently for each pair of neurons. This is important for a number of reasons. For example a neuron $i$ might have an activity that is moderately similar to two other neurons, $j$ and $k$, while at the same time $j$ and $k$ have little affinity to each other. Here is an example to illustrate this subtlety, drawn directly from our experience with these neural data.  Imagine a brief time window during which $i$'s activity decreases and soon after increases, whereas $j$ decreases in tandem with $i$ but does not increase afterwards. In contrast, $k$ does not decrease initially, but does increase together with $i$. Then both $j$ and $k$ are somewhat similar to $i$, but not to each other.  This illustrates how, allowing the affinities between pairs of neurons to be treated independently, leads to a more appropriate measure of the likelihood of instantaneous interactions between them.
>
> **How sensitive are the results presented to the particular choice of affinity score? Would, e.g., a Spearman correlation between the two time series yield qualitatively similar results?**
>
> In prior work we tried many of the standard approaches and found that none of them agreed with our natural, intuitive assessment of how two traces should relate to one another. This qualitative mismatch was, in fact, the main motivation for our developing a novel affinity score. We mentioned, in Sec. 2, second paragraph, that previous studies were limited to traditional statistics and were not able to reveal the same relationships we found between neurons. Moreover, from Sec. 2.1, first paragraph: Affinities should thus be a function of time, rather than the more traditional method of computing a global-time measure such as correlation.
>
> Spearman correlation, in particular, computes an average across all values, so it could not be used to produce a continuous similarity value over time. Our affinity method, on the other hand, proposes a principled way to define local time windows in which the rates of change can be compared (in absolute value, as explained above).

---

> > ### Author Response · Authors · 2023-11-16
> >
> > **While the experimental data are a definite plus, it's always unclear how strongly they should be taken as validation of a particular data analysis method. In a strongly coupled network, ablating any one neuron is likely to have an effect, and it's not shown that the method proposed here would necessarily outperform others for selecting which perturbations to apply.**
> >
> > Although the network is indeed densely connected, it has been found experimentally over many previous studies that that does not actually imply that any neuron is likely to have an effect on a specific behavior such as salt avoidance. For example, ablation of most non-motor neurons in C. elegans, alone or together, has been shown to have no measurable effect. Moreover, many neurons have been tested for an effect in the salt circuit and failed to show anything at all. Here are several references that confirm this:
> >
> > [1] Bargmann CI, Horvitz HR. (1991) Chemosensory neurons with overlapping functions direct chemotaxis to multiple chemicals in C. elegans. Neuron, 7(5):729-42.
> > This seminal paper tested chemosensory neurons. See Table 1, and Figs. 3 & 4. The results show that by and large, of the seven sensory neurons tested, ASE is the main means of sensing salt. The other neurons have a mild if any effect, and appear to be only meaningful when coupled to ASE ablation.
> >
> > [2] Luo L, et al. (2014) Dynamic encoding of perception, memory, and movement in a C. elegans chemotaxis circuit. Neuron, 82(5):1115-28.
> > See Fig. 6: of the 4 interneurons tested, AIB is the only one showing an effect when killed.
> >
> > [3] Yan G, et al. (2017) Network control principles predict neuron function in the Caenorhabditis elegans connectome. Nature, 550(7677):519-523.
> > Here the authors use control theory to predict which motor neurons will have a phenotypic effect when ablated. Their results again show that killing even some motor neurons does not yield a significant effect.
> >
> > [4] Avery, L., & Horvitz, H. R. (1989). Pharyngeal pumping continues after laser killing of the pharyngeal nervous system of C. elegans. Neuron, 3(4), 473–485.
> > See Table 1: Killing most of the pharyngeal neurons doesn’t have much of an effect.
> >
> > **I might have missed this, but how did the authors decide how many tensor components to retain?**
> >
> > Our logic for selecting the number of tensor components, R, was to use as many as possible to minimize reconstruction error, provided the results across multiple random initializations remained stable (i.e., small variance). Small reconstruction error suggests a faithful representation, and small variability guarantees that the retained components are robust. Based on the plots now included in Appendix A.4 (see updated manuscript), the error variability (std. dev. across 15 runs) reaches a minimum when R~14--15, then increases sharply for R > 15. We therefore selected R=15. Panel (b) shows that the actual distributions of error for each choice of R are clearly separate until R=15, and starting at R=16 they begin to mix with other values of R, which further strengthens our confidence in our choice for the number of tensor components.

---

> > ### Comment · Reviewer_nAoq · 2023-11-20
> >
> > > **The tensor factorization will tend to treat affinities independently of one another, though the N(N−1)/2 affinities result from only N neuronal time series.**
> >
> > Yes, this is what I was asking about, but I'm still a bit confused as to why treating these affinities as independent -- which ignores  some key structure in the data that generated them -- should be preferred. Clearly, all the information is extracted from the underlying data, and the authors have shown that the empirical performance is good. I'm just not entirely sure why, but this is a minor point.
> >
> > I appreciate the authors' replies to my other questions. I think this is nice work that still seems (to me) fairly tightly tuned to this particular application, so I will be maintaining my score.

---

### Official Review · Reviewer_6VUY · 2023-11-01

**Soundness:** 3 good
**Presentation:** 3 good
**Contribution:** 2 fair
**Rating:** 6
**Confidence:** 4

**Summary:**

This study presents an unsupervised method to identify the dynamic interactions between neurons. The approach has two main steps: first, based on calcium activity, the neural traces are organized to identify groups of neurons likely interacting over specific time intervals. Then, a generative model is applied to detect weighted communities in the functional patterns.

**Strengths:**

The paper is well-written and easy to follow. It is well-organized with illustrative figures. Also, the proposed method is simple, yet potentially effective for different tasks and domains. Moreover, compared to existing studies that use step-by-step statistical methods, this paper designs a method that allows considering the full system of similarities and animals across time. Also, presenting extensive experimental results supports the claim and the potential of the approach.

**Weaknesses:**

1. The main weakness of this paper is the lack of novelty in the model design. In fact, the proposed approach is a simple combination of existing methods and I cannot see a novel insight or a novel contribution from the machine learning side.

2. It would be better to include more related baselines. In the literature, there are several learning methods that learn the structure of the brain networks. Based on the current set of baselines, the proposed method shows superior performance, but existing baselines are general graph learning methods and do not use special properties of the brain. Therefore, I suggest adding additional brain network-based graph learning approaches as baselines.

3. There is a lack of discussion and experimental results about the scalability of the method and its efficiency. It would be great if the authors could provide more information about the efficiency of the method.

**Questions:**

Please see the Weaknesses.

---

> ### Author Response · Authors · 2023-11-16
>
> We thank the reviewer for their time and for their insightful comments and suggestions for the paper.
>
> **The main weakness of this paper is the lack of novelty in the model design. In fact, the proposed approach is a simple combination of existing methods and I cannot see a novel insight or a novel contribution from the machine learning side.**
>
>
> Although our algorithm does employ two well-understood components, we stress the novel insight is in how they are used. While non-negative tensor factorization (NTF) has been used in analyzing neuronal responses (e.g., [1], [2]), no one has previously used it successfully to define neural communities. Our critical contribution is the tensor dimension: *we seek to organize the affinities between pairs of neurons directly*. It is these neural pairs that lead, through the pipeline, to dynamic neural communities. Attempts to build tensor dimensions on the neurons directly could not accomplish this level of community organization.
>
> To elaborate, instead of implicitly computing similarities between the traces within the NTF computation (which, under the hood, would cause their comparison to be based entirely on linear dot products), we explicitly pre-compute our nonlinear differential affinities. If we had worked with the activities directly it would have yielded tensor factorizations based on how similar the individual traces were; our approach goes one order higher and organizes the affinities directly. Working with individual neurons would produce factors based on how similar the raw traces were.
>
> There is another advantage to our approach to teasing apart the functional connectome in neural networks. We use a novel nonlinear differential affinity, which enables (i) a differential comparison between neural activity traces that is not dependent on the actual calcium fluorescence values, and (ii)  accounts for both positive and negative relationships. Other recent studies have been limited to using traditional correlation-like measures of similarity between activity traces, and failed to find the neural interactions that we reveal in this paper (e.g., [3], [4], [5], [6]).
> Because most of the affinity computation is developed in Appendix A.1, we are adding a sentence to the main text of the manuscript to highlight this novelty.
>
> [1] Williams, A. H., et al. (2018). Unsupervised discovery of demixed, low-dimensional neural dynamics across multiple timescales through tensor component analysis. Neuron, 98(6), 1099-1115.\
> [2] Dyballa, L., et al. (2023). Population encoding of stimulus features along the visual hierarchy. bioRxiv.\
> [3] Susoy, V., et al. (2021). Natural sensory context drives diverse brain-wide activity during C. elegans mating. Cell, 184(20), 5122-5137.\
> [4] Randi, F., et al. (2022). Neural Signal Propagation Atlas of C. Elegans. arXiv preprint arXiv:2208.04790.\
> [5] Yemini, E., et al. (2021). NeuroPAL: a multicolor atlas for whole-brain neuronal identification in C. elegans. Cell, 184(1), 272-288.\
> [6] Kato, S., et al. (2015). Global brain dynamics embed the motor command sequence of Caenorhabditis elegans. Cell, 163(3), 656-669.\
>
> **It would be great if the authors could provide more information about the efficiency of the method.**
>
> Thank you for pointing this out. We realize this information is highly relevant in a context where experiments can collect increasing amounts of data. One of the reasons behind our choice of algorithm for NTF (HALS) was precisely its efficiency; it has been thoroughly benchmarked against other popular algorithms in [1], using a variety of real-world datasets.
>
> We have modified the manuscript to include this information (Sec. 2.2, 4th paragraph). For reference, our own experiments with running HALS on our dataset took about 35 s on a 3.1 GHz Dual-Core Intel Core i5 MaxBookPro laptop with 16GB of memory.
>
> [1] A.-H. Phan and A. Cichocki. Multi-way nonnegative tensor factorization using fast hierarchical alternating least squares algorithm (HALS). In Proc. of The 2008 International Symposium on Nonlinear Theory and its Applications, Budapest, Hungary, 2008.

---

> > ### Author Response · Authors · 2023-11-16
> >
> > **It would be better to include more related baselines. In the literature, there are several learning methods that learn the structure of the brain networks.**
> >
> > Although we didn’t develop this point in the manuscript explicitly, hoping to address the extended audience for  ICLR, we did examine methods that have been used on neuroscience data. A popular reference in the brain network literature [1] recommends two of the approaches used in our benchmark to perform network detection: modularity maximization (on which the Louvain and Combo methods are based), and statistical block models (on which the NWSBM method is based). A third approach recommended was the Infomax algorithm; although we did test it, Infomax performed well below the others so we did not include it.
> >
> > Another popular survey exploring community structure specific to brain networks [2] also recommends using modularity-based algorithms. Several studies [3,4,5] utilize a variation of the popular Louvain method (used in our benchmark) for dynamic community detection.
> >
> > We also found a study that utilized a method based on asymptotical surprise [6], which we now have added to our benchmark. The manuscript has been updated to reflect this (Sec. 3.3 and Table 1).
> >
> > [1] Sporns, O., & Betzel, R. F. (2016). Modular brain networks. Annual review of psychology, 67, 613-640.\
> > [2] Garcia, J. O., Ashourvan, A., Muldoon, S., Vettel, J. M., & Bassett, D. S. (2018). Applications of community detection techniques to brain graphs: Algorithmic considerations and implications for neural function. Proceedings of the IEEE, 106(5), 846-867.\
> > [3] Bassett, D. S., et al. (2013). Robust detection of dynamic community structure in networks. Chaos: An Interdisciplinary Journal of Nonlinear Science, 23(1).\
> > [4] Ashourvan, A., Telesford, Q. K., Verstynen, T., Vettel, J. M., & Bassett, D. S. (2019). Multi-scale detection of hierarchical community architecture in structural and functional brain networks. Plos one, 14(5), e0215520.\
> > [5] Mueller, J. M., et al. (2021). Dynamic community detection reveals transient reorganization of functional brain networks across a female menstrual cycle. Network neuroscience (Cambridge, Mass.), 5(1), 125–144.\
> > [6] Nicolini, C., & Bifone, A. (2016). Modular structure of brain functional networks: breaking the resolution limit by Surprise. Scientific reports, 6(1), 19250.

---

> ### Comment · Reviewer_6VUY · 2023-11-21
>
> I thank the authors for the new results and for answering my questions. I have raised my score to 6.
>
>
> While I appreciate the contributions of the paper from the neuroimaging and application side, I still believe that the novelty from the machine learning side is limited, which is the reason that I didn’t raise my score to 8.

---

### Official Review · Reviewer_vGem · 2023-11-09

**Soundness:** 4 excellent
**Presentation:** 4 excellent
**Contribution:** 3 good
**Rating:** 8
**Confidence:** 5

**Summary:**

The paper proposes a novel analysis method to infer time-varying functional connectomes from neuronal data in general and calcium imaging data in C. elegans in particular. The method is based on a three-step procedure. First, time-varying affinities between pairs of neurons are computed based on concurrent changes of the neuronal signal. Second, a non-negative tensor decomposition is employed to identify neuronal-temporal motifs of affinity across animals. Thirdly, community structure is inferred from the motifs using a stochastic block model. Taken together, these steps enable an interesting visualization of dynamical functional connectomes. The authors apply their method to experimental data recorded in C. elegans during a stimulus avoidance / attraction paradigm and identify a neuron previously not implicated in salt sensing. They then experimentally test its role by exposing worms with / without the neuron silenced to a salt stimulus and find that indeed the worms respond behaviorally as predicted.

**Strengths:**

The authors identify and address a highly relevant problem, i.e., the issue that relationships between neurons are highly dynamic yet few algorithms are able to infer dynamical functional connectomes. Their new methods enables a new visualization of this complex, high-dimensional data and can be used to derive experimentally testable predictions on the time-dependent involvement of neurons in behaviorally relevant neuronal ensembles. The authors further test and validate one specific hypothesis experimentally in a behavioral experiment. Another strength is that the manuscript is very well written and easy to read.

**Weaknesses:**

Some of the algorithmic choices appear rather ad-hoc without a rigorous theoretical or neurophysiological justification. In particular, it is unclear to me why the problems in constructing a time-varying similarity measure, that the authors discuss in the second paragraph of Section 2.1, does not also apply to the derivatives of the calcium traces? Since the derivatives represent the influx/outflux of calcium, and are thus likely a better representation of the neurons' firing rates, I would think that similar problems persist? Also, it is not clear to me why the local differential affinities should be non-negative? One could argue that two neurons also form a network if one inhibits the other, which in my understanding would lead to a negative affinity?

A further (and significant) weakness is that no link to code is provided in the manuscript. I believe that making all code publicly available is absolutely essential for reproducibility.

**Questions:**

My most relevant question is regarding code availability -- why has the code not been made available, and how do the authors intend to remedy that situation?

Further questions are minor ones:

* Since the affinity matrices are symmetric, vectorizing these (and using Euclidean norms) does not seem to be the right choice here? Have you looked into proper distance metrics for symmetric matrices [1]?
* What does "CP" stand for on page 4? That abbreviation is not introduced?
* Why was the set of neurons restricted to sensory and inter-neurons? Did the results change when using all neurons?
* Would the method also work on other neuronal data modalities, e.g., spiking data?

1. Vemulapalli, Raviteja, and David W. Jacobs. "Riemannian metric learning for symmetric positive definite matrices." arXiv preprint arXiv:1501.02393 (2015).

---

> ### Author Response · Authors · 2023-11-16
>
> We appreciate the attentive reading, your comments, questions, and suggestions.
>
> **why has the code not been made available, and how do the authors intend to remedy that situation?**
>
> We agree with the reviewer that the code should be fully available. Our plan is to make it available on github, so it may become a standard technique in computational neuroscience; we did not include a link due to the anonymity clause. To illustrate our intention, we now attach a zip file to the submission containing the source code for our method.
>
> **it is unclear to me why the problems in constructing a time-varying similarity measure (...) does not also apply to the derivatives of the calcium traces? Since the derivatives represent the influx/outflux of calcium (...) I would think that similar problems persist?**
>
> You are correct in that if the derivatives were to be compared using standard approaches such as Euclidean distance or cosine similarity, the same issues would persist. However, as discussed in detail in Appendix A.1, the comparison between derivatives is highly nonlinear:
>
> First, discrete time windows within which to compute local affinities are restricted to periods during which a neuron's derivative has constant sign, i.e., periods of monotonic influx or outflux of calcium. Then, the affinity is computed as the normalized intersection between the integrated calcium influx/outflux of each neuron (areas under the neurons' derivative curves). This approach acts to effectively eliminate the pitfalls presented in the second paragraph of Sec. 2.1, as has been empirically demonstrated.
>
> **it is not clear to me why the local differential affinities should be non-negative? One could argue that two neurons also form a network if one inhibits the other, which in my understanding would lead to a negative affinity?**
>
> Yes: negative affinities are not only possible, they indeed frequently occur in our data. That is precisely the reason why we use absolute values: our goal is for the affinity to represent a (non-negative) “probability” that two neurons interact, regardless of the qualitative nature of that interaction (excitation or inhibition).
>
> Although the affinity derivation is developed in detail in Appendix A.1, based on your comments we now realize that this particular feature of our method should be more clearly stated in the main text. So we have updated the manuscript to include in Sec. 2.1, 4th paragraph (new text in bold):
>
> *"Affinities show how similar two curves are, locally, in terms of their absolute derivatives, __which is motivated by the fact that two neurons with very similar but opposite sign derivatives are still likely to be interacting by means of some inhibitory mechanism.__ As a result, they can be interpreted as how likely it is for two neurons to be interacting"*
>
>
> **Since the affinity matrices are symmetric, vectorizing these (and using Euclidean norms) does not seem to be the right choice here?**
>
> Thanks for the suggestion as well as the interesting reference. Note, however, that in tensor factorization we do not compare affinity matrices as a whole. Rather, because tensor factorization is based on an alternating least-squares (ALS) approach, at each iteration the algorithm solves a problem of the form:
>
> $\mathbf{A} \leftarrow \textup{arg} \min_{\mathbf{A}} \left \|\mathbf{X}_{(1)} - (\mathbf{C}\odot \mathbf{B}) \mathbf{A}^{T}  \right \|$
>
> $\mathbf{B} \leftarrow \textup{arg} \min_{\mathbf{B}} \left \|\mathbf{X}_{(2)} - (\mathbf{C}\odot \mathbf{A}) \mathbf{B}^{T}  \right \|$
>
> $\mathbf{C} \leftarrow \textup{arg} \min_{\mathbf{C}} \left \|\mathbf{X}_{(3)} - (\mathbf{B}\odot \mathbf{A}) \mathbf{C}^{T}  \right \|$
>
> where $\mathbf{X}_{(i)}$ is the mode-$i$ matricization of the tensor $\mathbf{X}$ (following the notation from [1]). Therefore, the Euclidean distance here is never actually computed with respect to any of the symmetric affinity matrices (slices of $\mathbf{X}$), but rather to non-square, matricized versions of the entire tensor. It would therefore be non-trivial to adapt these techniques to our case. Of course, it is possible in the future that we will find an alternative metric which is better, but for now the standard approach suffices.
>
> [1] Rabanser, S., Shchur, O., & Günnemann, S. (2017). Introduction to tensor decompositions and their applications in machine learning. arXiv preprint arXiv:1711.10781.
>
> [1] Kolda, T. G., & Bader, B. W. (2009). Tensor decompositions and applications. SIAM review, 51(3), 455-500.

---

> > ### Author Response · Authors · 2023-11-16
> >
> > **What does "CP" stand for on page 4? That abbreviation is not introduced?**
> >
> > Thanks for noticing that. CP stands for Canonical Polyadic decomposition [1]. We have now edited the manuscript to introduce this abbreviation.
> >
> > **Why was the set of neurons restricted to sensory and inter-neurons? Did the results change when using all neurons?**
> >
> > Motor neurons (the only type we left out) control musculature to coordinate locomotion. Silencing motor neurons may cause locomotory behavioral effects irrespective of stimulus. Thus we would be gaming the system by predicting that they would have a phenotype for salt responses – they will have an effect for just about anything; this has been demonstrated experimentally [1]. Therefore, we left them out of our analysis. Of course, from the perspective of the algorithm, there is nothing that prevents one from using all neurons in another study.
> >
> > [1] Kato, S., et al. (2015). Global brain dynamics embed the motor command sequence of Caenorhabditis elegans. Cell, 163(3), 656-669.
> >
> >
> > **Would the method also work on other neuronal data modalities, e.g., spiking data?**
> >
> > Yes. In fact, we are also applying the same approach using data from electrophysiology, as well as data from artificial neural networks. So it is certainly a method with wide applicability.

---

> > ### Comment · Reviewer_vGem · 2023-11-22
> >
> > Thanks for the clarifications. They have reinforced my assessment that this is very solid and interesting work that is likely to be influential in neurobiology. I do acknowledge the criticism of one of the other reviewers that the ML/AI novelty is limited. But I do think that (in line with the Call for Papers) "applications to neuroscience & cognitive science" have a place at ICLR.

---

### Author Response · Authors · 2023-11-16
**Summary of revisions to manuscript**

-Clarified the handling of negative affinities within the main text.\
-Added comment on computational efficiency.\
-Expanded benchmark to include an additional method from the brain networks literature.\
-Added new Appendix section with plots illustrating the selection of number of tensor components.\
-Added zip file containing the source code used in our methods.\
-Fixed a mistake in Fig. A1 where one adjacency matrix was repeated.\
-Fixed typos and additional minor details.

---

### Meta-Review · Area_Chair_U1SD · 2023-12-05

**Metareview:**

The paper proposes an analysis method to infer time-varying functional/effective connectivities from neuronal data, focusing on calcium imaging data in C. elegans in particular. The method is very practically oriented, and based on a three-step procedure [rather than an end-to-end optimised approach], including a final step which fits a stochastic block model to the data. The reviewers generally agreed that this procedure steps enables an interesting visualization of dynamical functional connectomes.

**Justification For Why Not Higher Score:**

see above

**Justification For Why Not Lower Score:**

see above

---

### Decision · Program_Chairs · 2024-01-16

Accept (poster)